# Using an Artificial Intelligence Approach to Predict the Adverse Effects and Prognosis of Tuberculosis

**DOI:** 10.3390/diagnostics13061075

**Published:** 2023-03-13

**Authors:** Kuang-Ming Liao, Chung-Feng Liu, Chia-Jung Chen, Jia-Yih Feng, Chin-Chung Shu, Yu-Shan Ma

**Affiliations:** 1Department of Internal Medicine, Chi Mei Medical Center, Chiali, Tainan 722013, Taiwan; 2Department of Medical Research, Chi Mei Medical Center, Tainan 710402, Taiwan; yushan.ma.72@gmail.com; 3Department of Information Systems, Chi Mei Medical Center, Tainan 710402, Taiwan; 4Department of Chest Medicine, Taipei Veterans General Hospital, Taipei 112201, Taiwan; 5School of Medicine, National Yang-Ming University, Taipei 112304, Taiwan; 6Department of Internal Medicine, National Taiwan University Hospital, Taipei 100225, Taiwan; 7College of Medicine, National Taiwan University, Taipei 100233, Taiwan

**Keywords:** tuberculosis, acute hepatitis, respiratory failure, mortality, artificial intelligence, machine learning

## Abstract

Background: Tuberculosis (TB) is one of the leading causes of death worldwide and a major cause of ill health. Without treatment, the mortality rate of TB is approximately 50%; with treatment, most patients with TB can be cured. However, anti-TB drug treatments may result in many adverse effects. Therefore, it is important to detect and predict these adverse effects early. Our study aimed to build models using an artificial intelligence/machine learning approach to predict acute hepatitis, acute respiratory failure, and mortality after TB treatment. Materials and Methods: Adult patients (age ≥ 20 years) who had a TB diagnosis and received treatment from January 2004 to December 2021 were enrolled in the present study. Thirty-six feature variables were used to develop the predictive models with AI. The data were randomly stratified into a training dataset for model building (70%) and a testing dataset for model validation (30%). These algorithms included XGBoost, random forest, MLP, light GBM, logistic regression, and SVM. Results: A total of 2248 TB patients in Chi Mei Medical Center were included in the study; 71.7% were males, and the other 28.3% were females. The mean age was 67.7 ± 16.4 years. The results showed that our models using the six AI algorithms all had a high area under the receiver operating characteristic curve (AUC) in predicting acute hepatitis, respiratory failure, and mortality, and the AUCs ranged from 0.920 to 0.766, 0.884 to 0.797, and 0.834 to 0.737, respectively. Conclusions: Our AI models were good predictors and can provide clinicians with a valuable tool to detect the adverse prognosis in TB patients early.

## 1. Introduction

Tuberculosis (TB) is an infectious disease that spreads directly from one person to another and is a major cause of morbidity and mortality worldwide. It is also one of the leading causes of death from a single infectious disease and is more prevalent than human immunodeficiency virus/acquired immunodeficiency syndrome (HIV/AIDS) worldwide [1].

Patients infected with TB can be effectively treated with anti-TB medication, and the drug regimen, dosage, and length of treatment period depend on whether it is a drug-resistant strain, what comorbidities are present (diabetes, HIV, liver disease, renal disease, etc.), and where is the infection located in the body [2]. Most tuberculosis medications can be toxic to the liver and have the adverse effect of hepatitis. Therefore, when patients take these anti-TB medications, physicians need to monitor the patient’s liver enzymes and be aware of the risk of hepatitis. For example, Ramappa and Aithal [3] found that the TB medication that can cause hepatitis included isoniazid, rifampicin, and pyrazinamide. Patients with TB who develop hepatitis during the treatment may need to change TB medications if the hepatitis is severe. On the other hand, Elhidsi et al. [4] found that most patients with TB with acute respiratory failure were newly diagnosed patients, and had advanced lesions and hypoxemic type respiratory failure. The independent risk factors of in-hospital mortality were severe hypoxemia and kidney injury. Another study [5] showed that advanced age and presence of shock unrelated to sepsis were independently associated with mortality after multivariate analysis.

Artificial intelligence (AI)-based computer programs can assist hospitals in reading chest radiographs in a timely fashion, these programs perform similarly to expert physicians and radiologists with high sensitivity in detecting TB disease to determine which patients need further examination [6].

Recently, most studies have used AI and machine learning (ML) models to diagnose TB and explore the data characteristics and features used for algorithm accuracy [7]. Limited studies have focused on predicting adverse outcomes such as mortality and treatment failure [8,9,10]. Our study aimed to use the AI/ML model to detect hepatitis, respiratory failure, and mortality early in patients with TB after receiving anti-TB medications.

## 2. Methods

### 2.1. Study Design, Setting, and Samples

We retrospectively collected the data of first-visit patients with TB, and did not include therapy-refractory TB with second-line anti-TB drugs, from the three hospitals of Chi Mei Medical Group in Taiwan (1 medical center, 1 regional hospital, and 1 district hospital) from 1 January 2004 to 31 December 2021, and the patients had the diagnosis codes of TB (ICD-9: 010, 011, 012, 505, 647.3, 013, 014, 015, 016, 017, 018, 771.2 or ICD-10: J A15, J65, O98.0, A17, A18, A19, P37.0). Data from patients under 20 years old at the time of diagnosis, those with nontuberculous mycobacteria (NTM), and missing values were excluded. Overall, 4018 raw cases were included in the study (Figure 1).

### 2.2. Feature and Outcome Variables

We chose three outcome variables for the prediction models: (1) acute hepatitis, (2) acute respiratory failure, and (3) all-cause mortality during treatment.

The death certificate data were obtained through a formal application to Taiwan’s Health and Welfare Data Science Center.

The diagnosis of acute hepatitis must meet at least one of the following criteria:

Condition 1: The initial alanine aminotransferase (ALT, GPT) or aspartate aminotransferase (AST, GOT) is three times (or higher) the upper limit of the normal range during the treatment period.

Condition 2: The initial ALT (GPT) or AST (GOT) is more than twice the original value during the treatment period.

Condition 3: The total bilirubin (T-Bil) is >3 mg/dL during treatment.

Condition 4: The T-Bil baseline (the latest one before treatment) is abnormal (>1.2 mg/dL).

The diagnosis of acute respiratory failure must meet any of the following disease codes: ICD-9: 518.81, 518.84; ICD-10: J96.00, J96.01, J96.02, J96.20, J96.21, J96.22, J96.9, J96.90, J96.91, J96.92.

The treatment period is from the date of starting the TB medication to the date of the completion of the TB treatment. The normal value of ALT (GPT) is 41 U/L; the normal value of AST (GOT) is 31 U/L; and the normal value of T-Bil is 1.2 mg/dL in our laboratory.

Furthermore, we chose 36 feature variables, based on literature evidence and clinical experience, for these models. The features included sex, age, TB type (extra-pulmonary TB, any clinically diagnosed or bacteriologically confirmed case of TB affecting organs other than the lungs; intra-pulmonary TB, clinically diagnosed or bacteriologically confirmed case of TB involving lungs; both (intra-pulmonary TB and extra-pulmonary TB)), and disease history (diabetes mellitus (DM), hypertension, dyslipidemia, end-stage renal disease (ESRD), cerebrovascular accident (CVA), dementia, congestive heart failure (CHF), chronic obstructive pulmonary disease (COPD), asthma, malignancy, autoimmune disease, liver cirrhosis, old TB, hepatitis, pleural effusion). We also recorded all TB medication, including rifater, rifinah (150/101 mg), mycobutin (151 mg), rifinah (300/151 mg) isoniazid (101 mg), E-butol (401 mg), pyrazinamide (501 mg), rifampicin (151 mg). Finally, laboratory data included hepatitis B surface antigen (HBsAg), anti-hepatitis C virus (anti-HCV), white blood cell (WBC) count, hemoglobulin (Hb), platelet count, blood urea nitrogen (BUN), creatinine, AST (GOT), ALT (GPT), and T-Bil.

### 2.3. Model Building and Evaluation

We used all the variables to build the prediction models to maximize model performance without performing any feature selection preprocessing. The data were randomly stratified into a training dataset (70%) and a testing dataset (30%). The SMOTE method (synthetic minority oversampling technique) [11] was used to fix the data imbalance due to the fewer related positive classes (outcomes to be predicted, such as mortality) in the training dataset. The model of each outcome was built with 6 machine learning algorithms, including (1) multilayer perceptron (MLP), (2) LightGBM, (3) random forest, (4) XGBoost, (5) logistic regression, and (6) support vector machine (SVM).

We used a grid search with 10-fold cross-validation to build the best models based on the training dataset. We then used the testing dataset to evaluate the models with the performance indicators of accuracy, sensitivity, specificity, and AUC (area under the receiver operating characteristic curve).

## 3. Results

Finally, after removing data with missing values, 2248 patients were enrolled for model building. The data distribution and significance are summarized in Table 1, showing that the mean age of the patients was 67.7 years old, 71.7% were males, and 28.3% were females. According to the Spearman correlation analysis (Figure 2), the most relevant features to acute hepatitis were S-GOT, S-GPT, and T-Bil before hepatitis; those to acute respiratory failure were WBC, BUN, and age; and those to mortality were BUN, WBC, and age.

We used six machine learning algorithms to build the three outcome-predictive models of acute hepatitis, acute respiratory failure, and mortality. The results showed that the MLP algorithm obtained the highest AUC value (0.834) for the mortality prediction model (see Table 2 and Figure 3), the random forest algorithm had the highest AUC value (0.884) for acute respiratory failure (see Table 3 and Figure 4), and the XGBoost algorithm had the highest AUC value (0.920) for acute hepatitis (see Table 4 and Figure 5).

## 4. Discussion

To our knowledge, this is the first study to use AI and ML models to early detect acute hepatitis, respiratory failure, and mortality simultaneously in patients with TB after receiving anti-TB medications.

Our study included common clinical information and demographic data, such as age, sex, WBC, Hb, platelet count, BUN, creatinine, AST (GOT), ALT (GPT), bilirubin, comorbidities, and TB medication, to predict acute hepatitis, respiratory failure, and mortality in patients with TB after receiving TB medication. We also comprehensively included comorbidities, such as diabetes, hypertension, dyslipidemia, ESRD, CVA, dementia, CHF, COPD, asthma, malignancy, autoimmune disease, HIV, history of liver cirrhosis, hepatitis, old TB, and presence of pleural effusion in the models. With soft computing techniques, electrical medical systems can retrieve this information, and a clinician is not required to survey and rearrange the examination data. Moreover, our study evaluated laboratory data and systemic diseases in predicting TB patients’ prognosis.

We compared previous related studies on predicting adverse outcomes of TB patients [8,9,10,12,13], and found that our predictive model was based on the literature and practical availability, and had excellent quality (AUCs: 0.834~0.920), which is quite worthy of being developed as a predictive tool to assist in clinical decision-making. We summarized the comparison of these works in Table 5.

### 4.1. Acute Hepatitis

Luo et al. [14] enrolled patients with active TB and latent TB infection in China based on multiple laboratory data and used different models established by ML for distinguishing the patient’s TB infection status. Nijiati et al. [15]. used a three-dimensional model to detect lung field regions in CT images and ML methods for classification and differentiating active/nonactive pulmonary TB. With AI assistance, radiologists working in this field can truly help potential patients. Another study [16] used AI for training models to interpret chest X-ray images and achieved high accuracy. These recent studies used AI and ML to detect TB early and did not mention how to detect hepatitis in patients with TB.

Risk factors for hepatitis after TB treatment have been assayed in an observational study [17]. Among the various risk factors assessed, extensive disease, old age, excessive alcohol use, and slow acetylator phenotype were risk factors for hepatitis in patients who received anti-TB drugs. A study enrolled 765 patients who received anti-TB treatment and found that the risk factors for hepatotoxicity included advanced age, female sex, extensive tuberculosis, and no alcohol consumption [18]. In our population, there was no significant difference in age between TB patients with and without acute hepatitis. Wang et al. showed that age and hepatitis B infection were important risk factors for hepatitis in patients with TB via a multiple logistic regression analysis [19]. Extra-pulmonary TB, advanced age, and comorbidities were found to be significant predictors of the development of hepatitis in studies using multivariable logistic regression analyses [20]. However, our study showed no significant difference between extra-pulmonary and intra-pulmonary TB among patients with acute hepatitis. Approximately 12% of the patients died after the development of anti-TB drug-induced hepatitis [21]. In our study, most of these factors and patient laboratory data that were taken before acute hepatitis had developed were included in the models. However, alcohol intake was not included in our study because alcohol intake was not recorded in our electrical medical record and because this was a retrospective study. From our data, we found that patients with TB and acute hepatitis had a high proportion history of hepatitis and liver cirrhosis. Furthermore, with the aid of these variables and ML, the XGBoost model still had a high accuracy of 0.868, a sensitivity of 77.9%, a specificity of 92.5%, and an AUC of 0.920. In addition to higher accuracy in detecting hepatitis during the TB treatment course, we can detect hepatitis early in patients with TB. With the aid of AI, physicians can be aware of the risk of hepatitis and more frequently and intensively monitor liver function before this adverse effect occurs.

### 4.2. Acute Respiratory Failure

Despite the availability of effective anti-TB medications, TB, as a cause of respiratory failure requiring mechanical ventilation, is often associated with acute respiratory distress syndrome, which leads to a high mortality rate [22]. A study enrolled 41 patients with TB in Taiwan from January 1996 to April 2001; a total of 27 died (65.9%) in the hospital, and 14 survived, with a (mean ± sd) of 40.7 ± 35.4 admission days before death. The mortality rate for the 180 day monitoring period was 79% [23]. The multivariate analysis found that old age, multiple organ failure, and shock unrelated to sepsis were related to poor outcomes [5]. Therefore, detecting patients with TB at risk of acute respiratory failure from complex diseases and patients with multiple comorbidities earlier is important for clinical care. We found that our patients with TB and respiratory failure requiring mechanical ventilation had a higher proportion of CVA, dementia, CHF, COPD, and TB present with pleural effusion. These baseline comorbidities may reflect the fragility in patients with TB, and the elderly with these comorbidities are more vulnerable to respiratory failure. TB effusion was a common condition, and treatment of TB with pleural effusion was the same as for pulmonary TB. Most TB patients with pleural effusion had a benign course after treatment with mild-to-no residual effects. There was scarce literature regarding TB with pleural-effusion-related acute respiratory failure. Further studies were required to answer if pleural effusion is meaningful to TB patients with acute respiratory failure.

The early signs of acute respiratory failure may be uncertain in some laboratory test results. Predictive models using AI that integrate and leverage multiple variable factors could help identify areas of uncertainty, and this identification would likely occur before any noticeable physical symptoms appear. By incorporating ML into laboratory data, routine data results can be merged into other relevant patient characteristics, such as age, sex, and comorbidities, for use within disease-specific AI models. By integrating information, patient characteristics, and laboratory data, there is a potential to generate acute respiratory failure patient probability scores to help alert clinicians. Our predictive random forest models’ accuracy, sensitivity, specificity, and AUC were 0.819, 0.812, 0.820, and 0.884, respectively. In collaboration with more patient information and healthcare institutions, ML and computerized reasoning can be used to develop AI-driven clinical decision support tools that can potentially aid clinicians in making prompt and correct decisions before TB patients experience acute respiratory failure.

### 4.3. Mortality

TB hurts the patients’ long-term survival rate even after successful treatment and decreases the survival rate in long-term follow-up, even after accounting for acute TB-related mortality [24]. The survival rate at 11 years was 70% after successful TB treatment, and the probability of survival was 46% in the age group of 55 years and older after 11 years of follow-up [25]. Another study also showed that mortality in the TB cohort was 2.3 times higher than in the general population after age matching. Most mortality occurred in the first year after completing treatment [26]. During our 17 year follow-up, we enrolled 2128 patients with TB, and there were 120 deaths during this period. The predictive model of MLP had an accuracy of 0.735, a sensitivity of 0.722, a specificity of 0.736, and an AUC of 0.834. Our data found that extra-pulmonary TB had a low risk of mortality, and comorbidities of hypertension, CVA, dementia, CHF, and COPD were associated with mortality among patients with TB compared with those without these comorbidities. A previous study showed advancing age and drug resistance were the features most associated with risk of death. In contrast, male sex, European origin, pulmonary site of TB infection, and previous history of anti-TB treatment were weaker predictors [27] but our data are inconsistent with their results. The median age in their study was 43 years, with 2% aged < 15 years and 24% aged ≥ 60 years, 5% of patients had multidrug resistance, and most cases were European (68%). The inconsistencies may result from differences between countries, presumably reflecting the differences in patient characteristics and drug susceptibility to TB.

These outcomes are important for patients with TB and their medical teams. Focusing on patient-centered care and the early prediction of adverse drug effects, respiratory failure, and mortality in patients receiving TB treatment could contribute to the optimal use of medical resources.

In addition to accurate and prompt diagnosis of TB, it is important to detect the possible risk of hepatitis in patients who receive TB treatment as early as possible so that the culprit medicine can be discontinued to improve the patient’s outcome.

Our study also has some limitations that need to be addressed and explored. First, our patients were from southern Taiwan and may have differed from TB patients in other countries. Our models were not representative of other countries. Further studies in other areas with more hospitals may be needed for more representative results. Second, alcohol use in TB patients was considered an important risk factor for hepatitis. The study is retrospective, and data on smoking and drinking variables are missing. Our study could not include this factor because we could not accurately obtain this information from electronic medical records. Third, the model created in the current study lacks TB drug susceptibility, and future studies should focus on drug susceptibility testing. Fourth, the mortality of patients with TB failed to determine the direct cause. Thus, our results may not be recommended for the general extrapolation population of patients with TB.

## 5. Conclusions

In conclusion, we created a model based on laboratory data and patient characteristics that has significant value in the early detection of hepatitis, respiratory failure, and mortality in patients with TB who received anti-TB treatment.

## Figures and Tables

**Figure 1 diagnostics-13-01075-f001:**
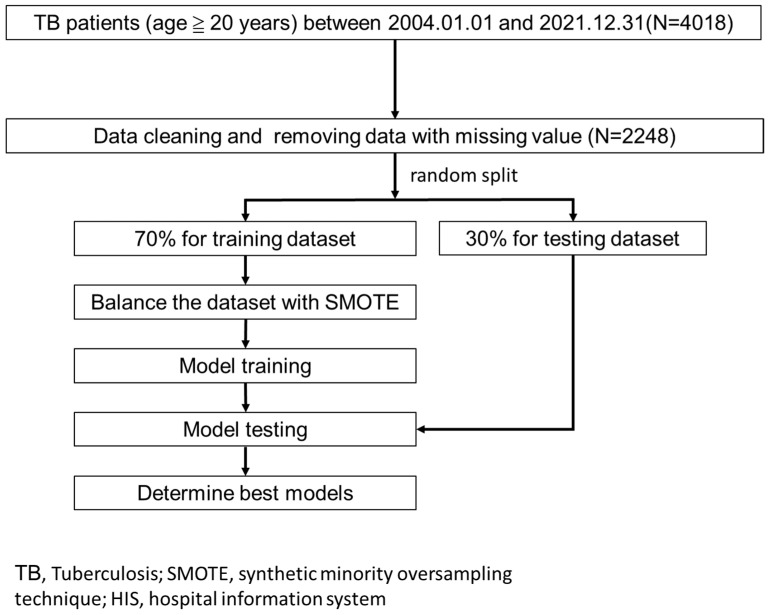
Research flow.

**Figure 2 diagnostics-13-01075-f002:**
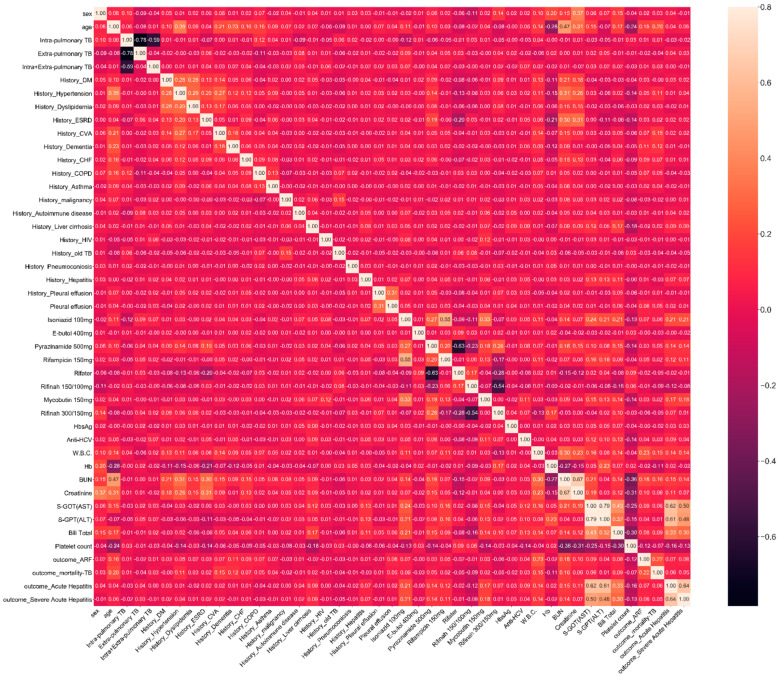
Spearman correlation.

**Figure 3 diagnostics-13-01075-f003:**
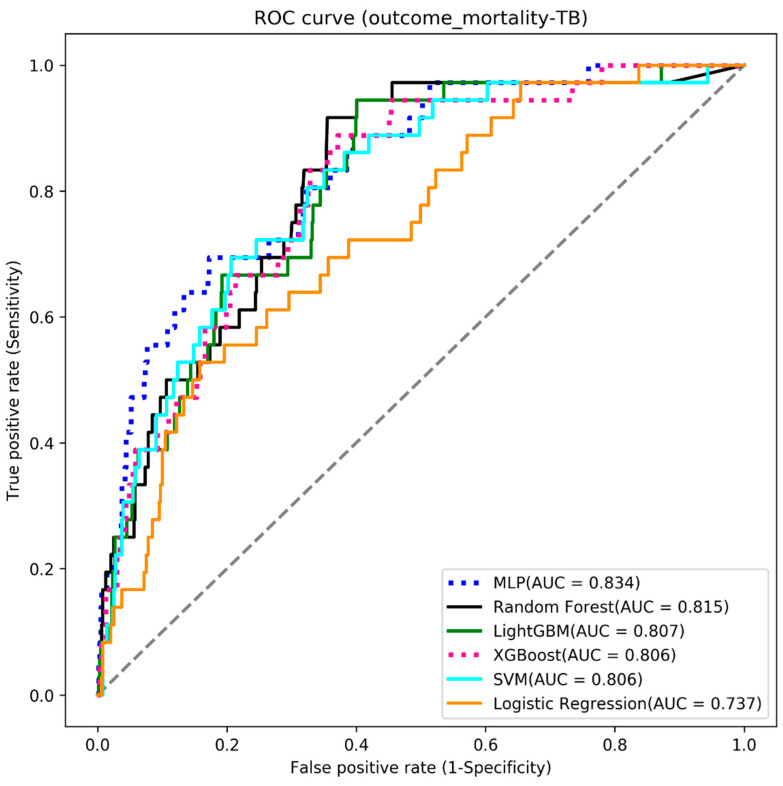
The ROC curve of the mortality models.

**Figure 4 diagnostics-13-01075-f004:**
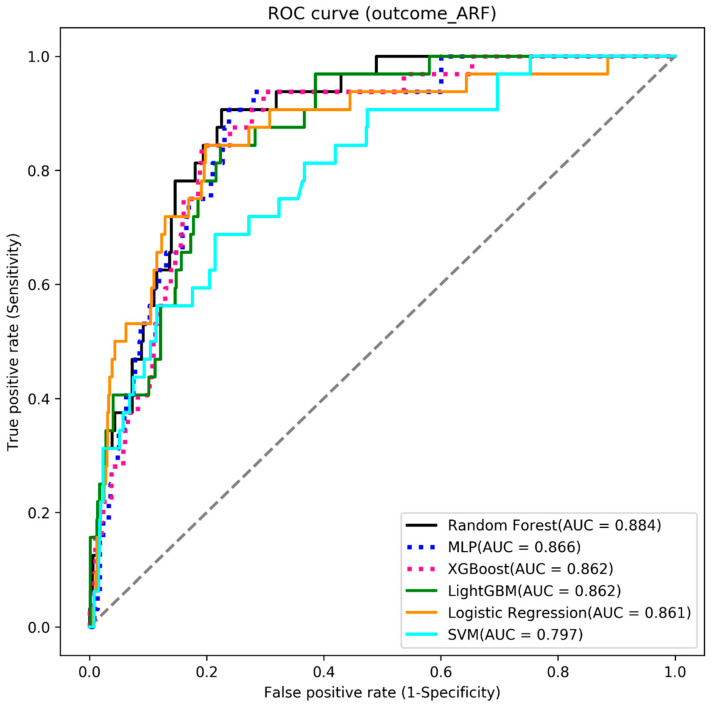
The ROC curve of the acute respiratory failure models.

**Figure 5 diagnostics-13-01075-f005:**
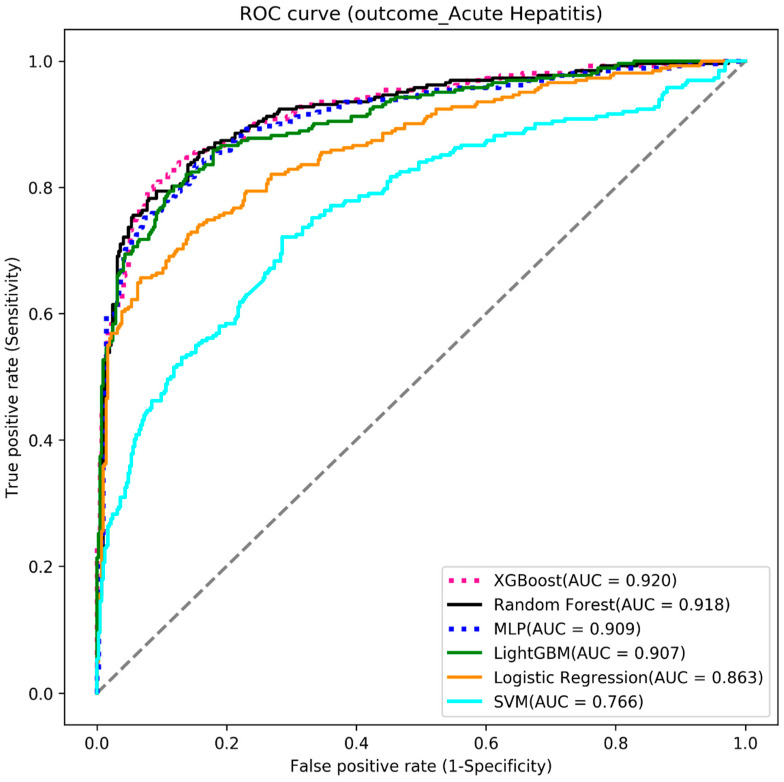
The ROC curve of the acute hepatitis models.

**Table 1 diagnostics-13-01075-t001:** Demographics.

Variable	Overall	Acute Hepatitis			Acute Respiratory Failure			Mortality		
		NO	Yes	*p* Value	NO	Yes	*p* Value	NO	Yes	*p* Value
Cases, n (%)	2248 (100.0)	1377 (61.3)	871 (38.7)		2141 (95.2)	107 (4.8)		2128 (94.7)	120 (5.3)	
sex, n (%)										
Female	637 (28.3)	420 (30.5)	217 (24.9)	0.005	611 (28.5)	26 (24.3)	0.401	608 (28.6)	29 (24.2)	0.348
Male	1611 (71.7)	957 (69.5)	654 (75.1)		1530 (71.5)	81 (75.7)		1520 (71.4)	91 (75.8)	
age, mean (SD)	67.7 (16.4)	67.2 (16.5)	68.5 (16.3)	0.071	67.1 (16.5)	79.9 (8.7)	<0.001	66.9 (16.4)	80.9 (9.9)	<0.001
TB_type, n (%)										
Extra-pulmonary	140 (6.2)	77 (5.6)	63 (7.2)	0.292	138 (6.4)	2 (1.9)	0.145	139 (6.5)	1 (0.8)	0.022
Intra-pulmonary	2025 (90.1)	1249 (90.7)	776 (89.1)		1925 (89.9)	100 (93.5)		1913 (89.9)	112 (93.3)	
Both (Intra + Extra)	83 (3.7)	51 (3.7)	32 (3.7)		78 (3.6)	5 (4.7)		76 (3.6)	7 (5.8)	
History_DM, n (%)	612 (27.2)	376 (27.3)	236 (27.1)	0.952	579 (27.0)	33 (30.8)	0.454	581 (27.3)	31 (25.8)	0.805
History_Hypertension, n (%)	780 (34.7)	485 (35.2)	295 (33.9)	0.541	733 (34.2)	47 (43.9)	0.051	713 (33.5)	67 (55.8)	<0.001
History_Dyslipidemia, n (%)	233 (10.4)	159 (11.5)	74 (8.5)	0.025	219 (10.2)	14 (13.1)	0.434	220 (10.3)	13 (10.8)	0.985
History_ESRD, n (%)	125 (5.6)	80 (5.8)	45 (5.2)	0.580	118 (5.5)	7 (6.5)	0.812	119 (5.6)	6 (5.0)	0.944
History_CVA, n (%)	278 (12.4)	170 (12.3)	108 (12.4)	0.978	256 (12.0)	22 (20.6)	0.013	237 (11.1)	41 (34.2)	<0.001
History_Dementia, n (%)	135 (6.0)	86 (6.2)	49 (5.6)	0.609	116 (5.4)	19 (17.8)	<0.001	117 (5.5)	18 (15.0)	<0.001
History_CHF, n (%)	161 (7.2)	98 (7.1)	63 (7.2)	0.984	142 (6.6)	19 (17.8)	<0.001	144 (6.8)	17 (14.2)	0.004
History_COPD, n (%)	693 (30.8)	440 (32.0)	253 (29.0)	0.159	638 (29.8)	55 (51.4)	<0.001	641 (30.1)	52 (43.3)	0.003
History_Asthma, n (%)	122 (5.4)	83 (6.0)	39 (4.5)	0.138	115 (5.4)	7 (6.5)	0.762	112 (5.3)	10 (8.3)	0.216
History_malignancy, n (%)	483 (21.5)	292 (21.2)	191 (21.9)	0.723	466 (21.8)	17 (15.9)	0.185	463 (21.8)	20 (16.7)	0.227
History_Autoimmune disease, n (%)	97 (4.3)	50 (3.6)	47 (5.4)	0.057	92 (4.3)	5 (4.7)	0.806	94 (4.4)	3 (2.5)	0.438
History_Liver cirrhosis, n (%)	68 (3.0)	25 (1.8)	43 (4.9)	<0.001	67 (3.1)	1 (0.9)	0.376	64 (3.0)	4 (3.3)	0.782
History_old TB, n (%)	172 (7.7)	108 (7.8)	64 (7.3)	0.727	167 (7.8)	5 (4.7)	0.317	168 (7.9)	4 (3.3)	0.098
History_Hepatitis, n (%)	100 (4.4)	44 (3.2)	56 (6.4)	<0.001	94 (4.4)	6 (5.6)	0.474	98 (4.6)	2 (1.7)	0.196
Pleural effusion, n (%)	20 (0.9)	10 (0.7)	10 (1.1)	0.420	16 (0.7)	4 (3.7)	0.013	17 (0.8)	3 (2.5)	0.087
Isoniazid 101 mg, n (%)	360 (16.0)	137 (9.9)	223 (25.6)	<0.001	333 (15.6)	27 (25.2)	0.011	333 (15.6)	27 (22.5)	0.062
E-butol 401 mg, n (%)	2236 (99.5)	1372 (99.6)	864 (99.2)	0.233	2130 (99.5)	106 (99.1)	0.444	2117 (99.5)	119 (99.2)	0.483
Pyrazinamide 501 mg, n (%)	699 (31.1)	378 (27.5)	321 (36.9)	<0.001	665 (31.1)	34 (31.8)	0.961	654 (30.7)	45 (37.5)	0.145
Rifampicin 151 mg, n (%)	416 (18.5)	190 (13.8)	226 (25.9)	<0.001	384 (17.9)	32 (29.9)	0.003	389 (18.3)	27 (22.5)	0.300
Rifater, n (%)	1877 (83.5)	1145 (83.2)	732 (84.0)	0.620	1788 (83.5)	89 (83.2)	0.966	1781 (83.7)	96 (80.0)	0.350
Rifinah 150/101 mg, n (%)	785 (34.9)	526 (38.2)	259 (29.7)	<0.001	742 (34.7)	43 (40.2)	0.286	758 (35.6)	27 (22.5)	0.005
Mycobutin 151 mg, n (%)	61 (2.7)	7 (0.5)	54 (6.2)	<0.001	56 (2.6)	5 (4.7)	0.211	56 (2.6)	5 (4.2)	0.376
Rifinah 300/151 mg, n (%)	1096 (48.8)	680 (49.4)	416 (47.8)	0.480	1067 (49.8)	29 (27.1)	<0.001	1055 (49.6)	41 (34.2)	0.001
HbsAg, n (%)										
Negative	2191 (97.5)	1347 (97.8)	844 (96.9)	0.224	2086 (97.4)	105 (98.1)	1.000	2072 (97.4)	119 (99.2)	0.366
Positive	57 (2.5)	30 (2.2)	27 (3.1)		55 (2.6)	2 (1.9)		56 (2.6)	1 (0.8)	
Anti-HCV, n (%)										
Negative	2166 (96.4)	1344 (97.6)	822 (94.4)	<0.001	2066 (96.5)	100 (93.5)	0.109	2053 (96.5)	113 (94.2)	0.203
Positive	82 (3.6)	33 (2.4)	49 (5.6)		75 (3.5)	7 (6.5)		75 (3.5)	7 (5.8)	
W.B.C., mean (SD)	10.5 (7.1)	9.6 (5.4)	12.0 (9.0)	<0.001	10.2 (6.9)	17.9 (8.3)	<0.001	10.3 (6.9)	15.1 (9.9)	<0.001
Hb, mean (SD)	12.9 (1.8)	12.9 (1.8)	13.0 (1.8)	0.409	12.9 (1.8)	12.8 (1.7)	0.426	13.0 (1.8)	12.1 (1.7)	<0.001
Platelet count, mean (SD)	201.0 (100.3)	213.8 (96.3)	180.8 (103.3)	<0.001	203.3 (99.5)	155.3 (106.0)	<0.001	203.0 (100.4)	167.1 (93.4)	<0.001
BUN, mean (SD)	28.5 (24.5)	25.8 (21.0)	32.7 (28.7)	<0.001	27.5 (23.2)	47.9 (37.8)	<0.001	27.6 (23.5)	44.8 (34.8)	<0.001
Creatinine, mean (SD)	1.7 (1.9)	1.6 (1.9)	1.9 (2.0)	0.007	1.7 (1.9)	2.2 (2.1)	0.030	1.7 (1.9)	2.1 (1.8)	0.018
AST (GOT), mean (SD)	116.9 (656.2)	41.1 (46.6)	236.6 (1041.8)	<0.001	112.0 (656.7)	215.3 (642.3)	0.107	113.6 (657.9)	175.5 (626.1)	0.295
ALT (GPT), mean (SD)	93.2 (242.2)	35.7 (50.1)	184.0 (366.1)	<0.001	90.9 (230.0)	138.4 (415.4)	0.243	91.7 (231.0)	118.6 (390.9)	0.457
Bili Total, mean (SD)	1.4 (2.7)	0.7 (0.4)	2.5 (4.1)	<0.001	1.4 (2.6)	1.8 (3.5)	0.226	1.4 (2.7)	1.8 (3.1)	0.128

SD: standard deviation, DM: diabetes mellitus, ESRD: end stage renal disease, CVA: cerebrovascular accident, CHF: congestive heart failure, COPD: chronic obstructive pulmonary disease, HBsAg: hepatitis B surface antigen, anti-HCV: anti-hepatitis C virus, WBC: white blood cell count, HB: hemoglobulin, BUN: blood urea nitrogen, AST (GOT): aspartate aminotransferase, ALT (GPT): alanine aminotransferase.

**Table 2 diagnostics-13-01075-t002:** Model results: mortality.

Algorithms	Accuracy	Sensitivity	Specificity	AUC
MLP	0.735	0.722	0.736	0.834
Random Forest	0.713	0.722	0.712	0.815
LightGBM	0.705	0.694	0.706	0.807
XGBoost	0.705	0.694	0.706	0.806
SVM	0.686	0.778	0.681	0.806
Logistic Regression	0.656	0.667	0.656	0.737

**Table 3 diagnostics-13-01075-t003:** Model results: acute respiratory failure.

Algorithms	Accuracy	Sensitivity	Specificity	AUC
Random Forest	0.819	0.812	0.820	0.884
MLP	0.812	0.812	0.790	0.866
XGBoost	0.812	0.812	0.812	0.862
LightGBM	0.812	0.750	0.815	0.862
Logistic Regression	0.806	0.781	0.807	0.861
SVM	0.727	0.719	0.728	0.797

**Table 4 diagnostics-13-01075-t004:** Model results: acute hepatitis.

Algorithms	Accuracy	Sensitivity	Specificity	AUC
XGBoost	0.868	0.779	0.925	0.920
Random forest	0.856	0.794	0.896	0.918
MLP	0.853	0.752	0.918	0.909
LightGBM	0.847	0.771	0.896	0.907
Logistic regression	0.776	0.779	0.775	0.863
SVM	0.717	0.721	0.714	0.766

**Table 5 diagnostics-13-01075-t005:** A comparison with past related studies.

Study	Our Study	[8]	[9]	[10]	[12]	[13]
Countries	Taiwan	Patients originated from India, Azerbaijan, Moldova, Georgia, Belarus, and Romania	Moldova	Azerbaijan, Belarus, Moldova, Georgia, Romania	Myanmar	Pakistan
Patient number	2248	1443	17,958	587	393	4213
Outcome	Acute hepatitis, acute respiratory failure, and mortality after TB treatment	Treatment failure, which is defined as failed in therapy or death	Cured, not cured, and died after 24 months following treatment initiation	Treatment failure, which we defined as failure of therapy or death	TB drug resistance	Patient will complete his treatment or not
Machine learning method	XGBoost, random forest, MLP, light GBM, logistic regression, and SVM.	Artificial Neural Network (ANN), Support Vector Machine (SVM), k-Nearest Neighbors (k-NN), and random forest (RF).	Random forest algorithm, support vector machine, penalized multinomial logistic regression models.	Stepwise forward selection, stepwise backward elimination, backward elimination and forward selection, Least Absolute Shrinkage and Selection Operator (LASSO) regression, random forest, and support vector machine (SVM) with linear kernel and polynomial kernel.	Genetic algorithm (GA) and support vector machine (SVM) model.	Artificial neural networks (ANNs), support vector machines (SVMs), random forest (RF).
Attribute data	Used 36 attributes including sex, age, TB type (intra-pulmonary TB or extra-pulmonary TB), disease history (diabetes mellitus, hypertension, dyslipidemia, end stage renal disease, cerebrovascular accident, dementia, congestive heart failure, chronic obstructive pulmonary disease, asthma, malignancy, autoimmune disease, liver cirrhosis, previous TB, hepatitis, pleural effusion, TB medication, and laboratory data.	Used 22 attributes including country, education level, sex, employment status, type of resistance, number of daily contacts, Body Mass Index (BMI), localization in the lung, number of X-rays, number of CT scans, dissemination, pleural effusion, pneumothorax, pleuritis, process extension, decrease in lung capacity, lung cavern, culture results, microscopy results, social risk factors (including smoking, alcoholism, ex-prisoner, Multi Drug-resistant patients etc.), and drug regimen.	Used 112 attributes including baseline covariates: gender, microbiological data, age of onset, TB group, direct smear test profile for second-line drugs; time-dependent covariates: smear, culture, direct smear test profile for first-line drugs	Used 28 attributes including country, age of onset, sex, education level, employment status, number of daily contacts, type of resistance, body mass index, localization in the lung, number of X-rays, number of CT scans, dissemination, size of the lung cavity, pleural involvement, imaging pattern, pneumothorax, pleuritis, nodal calcinosis, process extension, decrease in lung capacity, lung cavities, culture results, microscopy results, social risk factors, and drug regimen.	Used 35 attributes including sex, residence, occupation, marital, dwelling, drink, smoking, HIV, diabetes, alcohol, trips to traditional healer after TB positive, preferred health care provider to visit when sick, missing treatment in last 4 days, how often patient missed taking TB drugs private treatment type, whether patient takes traditional medicine, private doctor treatments in the past 24 months, traditional healer treatments in the past 24 months, medicine taken before TB diagnosis, and household income.	Used 52 attributes including demographics, screening, medical tests, Diagnosis, baseline treatment, and other variables related to TB treatment
Testing results	Area under the receiver operating characteristic curve in predicting acute hepatitis, respiratory failure, and mortality was 0.928, 0.884, and 0.834, respectively	Accuracy: 70–78%.	Sensitivity and positive predictive value increased to 0.84 and 0.88, respectively, for the not cured class.	Area under the receiver operating characteristic curve: 0.74.	SVM with GA is capable of achieving 67% of accuracy.	Accuracy: 76.01–76.32%.
Year	2022	2020	2022	2018	2021	2019

## Data Availability

The dataset used for this study is available on request to the corresponding author.

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
