# Peer review of "Using an Artificial Intelligence Approach to Predict the Adverse Effects and Prognosis of Tuberculosis"

_diagnostics, 2023, doi:10.3390/diagnostics13061075_

Round 1

Reviewer 1 Report

Liao et al. created an artificial intelligence (AI) or machine learning (ML) based model using laboratory data and patient characteristics for early detection of hepatitis, respiratory failure, and mortality in TB patients who received anti-TB treatment. 

Overall the model looks good and it can be used as an important tool to diagnose or predict consequences related to pulmonary tuberculosis. Although for confirmation laboratory testing is needed.

There are a few comments that need to be addressed-

Authors need to cite related references to Line 54 as there are several studies showing the role of AI in TB diagnosis.

The quality of Fig 1, Fig. 2, and Fig.  4 is poor. Please replace them with good-quality figures.

Reviewer 2 Report

Need major revision

Reviewer 3 Report

Dear Authors,

Please find my comments and queries, pertaining to your manuscript:

1.      Line 44: Please correct us such: what comorbidities are present (diabetes, HIV, liver disease, renal disease, etc.) and where is the infection located in the body.

2.      Line 47: Please correct us such: Patients with TB who develop hepatitis during the treatment may need to change TB medications if the hepatitis is serious.

3.      Lines 53-58: which TB medication can cause hepatitis, respiratory failure or death according to the present literature? Please provide some information about that.

4.      Line 61: Was the data collected of first-visit patients or first-diagnosis patients? Did any patients have a therapy-refractory TB with second-line anti-TB drugs?

5.      Line 72: how can you be sure that the mortality during the TB treatment was due to the medication and not due to other cause, the underlying disease for example?

6.      Line 75: Please correct us such: the diagnosis of acute hepatitis must meet at least one of the following criteria.

7.      Line 184: Please correct us such: Approximately 12% of the patients die after the development of anti-TB drug-induced hepatitis.

8.      Line 215: Please correct us such: The early signs of acute respiratory failure may be uncertain in some laboratory test results.

Best Regards

Round 2

Reviewer 2 Report

It can be accepted as such
